# Targeting PKCθ Promotes Satellite Cell Self-Renewal

**DOI:** 10.3390/ijms21072419

**Published:** 2020-03-31

**Authors:** Anna Benedetti, Piera Filomena Fiore, Luca Madaro, Biliana Lozanoska-Ochser, Marina Bouché

**Affiliations:** Dept AHFMO, University of Rome “La Sapienza”, Via A. Scarpa 14, 00161 Rome, Italy; anna.benedetti@uniroma1.it (A.B.); pierafilomena.fiore@opbg.net (P.F.F.); luca.madaro@uniroma1.it (L.M.); biliana.lozanoska-ochser@uniroma1.it (B.L.-O.)

**Keywords:** satellite cells, self-renewal, muscle regeneration, Protein kinase C θ

## Abstract

Skeletal muscle regeneration following injury depends on the ability of satellite cells (SCs) to proliferate, self-renew, and eventually differentiate. The factors that regulate the process of self-renewal are poorly understood. In this study we examined the role of PKCθ in SC self-renewal and differentiation. We show that PKCθ is expressed in SCs, and its active form is localized to the chromosomes, centrosomes, and midbody during mitosis. Lack of PKCθ promotes SC symmetric self-renewal division by regulating Pard3 polarity protein localization, without affecting the overall proliferation rate. Genetic ablation of PKCθ or its pharmacological inhibition in vivo did not affect SC number in healthy muscle. By contrast, after induction of muscle injury, lack or inhibition of PKCθ resulted in a significant expansion of the quiescent SC pool. Finally, we show that lack of PKCθ does not alter the inflammatory milieu after acute injury in muscle, suggesting that the enhanced self-renewal ability of SCs in PKCθ-/- mice is not due to an alteration in the inflammatory milieu. Together, these results suggest that PKCθ plays an important role in SC self-renewal by stimulating their expansion through symmetric division, and it may represent a promising target to manipulate satellite cell self-renewal in pathological conditions.

## 1. Introduction

Satellite cells (SCs) are the adult stem cells of skeletal muscle. They were first identified by Mauro in 1961who named them satellite cells because of their position between the sarcolemma and the basal lamina of the fiber [1]. Satellite cells are mitotically quiescent (in G0 phase) under steady state, and express the paired-box transcription factor Pax7 [2]. However, following muscle injury SCs enter the cell cycle, expand, and fuse together or with pre-existing myofibers to repair the muscle. During muscle regeneration some of the SCs undergo self-renewal to maintain the stem cell pool. Satellite cell self-renewal is achieved through the balance between asymmetric and symmetric division. Asymmetric division generates one stem cell and one committed progenitor, while symmetric division generates two identical cells that will either differentiate or self-renew [3,4]. The orientation of the mitotic spindle, the niche environment and the localization of the PAR polarity complex (PAR3-PAR6-aPKC), determine the type of division [5,6]. Thus, the asymmetric localization of the PAR complex in SCs drives the polarization of cell fate determinants and promotes asymmetric cell division by inducing MyoD expression in only one daughter cell [4].

Protein kinase C Theta (PKCθ) is a serine and threonine kinase belonging to the family of novel PKCs. It is mostly expressed in hematopoietic cells with high levels in T cells, where it regulates cell activation, proliferation, and survival [7,8]. In addition, PKCθ is also the PKC isoform predominantly expressed in skeletal muscle, where it regulates muscle development and homeostasis by promoting the initial myoblast fusion events. As a result, lack of PKCθ delays the early phase of muscle regeneration after muscle injury, but it does not compromise terminal differentiation [9].

We previously showed that knock-out of PKCθ or its inhibition in a mouse model of Duchenne Muscular Dystrophy (mdx mouse) reduces muscle damage, by modulating the immune cell infiltration. This beneficial effect was mainly due to the inhibition of PKCθ in T cells, which leads to reduced inflammation [10,11,12]. Interestingly, we also found that lack of PKCθ promotes muscle repair in dystrophic mice, even at advanced stages of the disease, supporting SCs survival and maintenance [13]. However, whether this effect was due to reduced inflammation or to a direct role of PKCθ on SCs activity, or both, was still unclear.

Here, we show that PKCθ plays a direct role in SCs activity by stimulating their symmetric division and promoting their self-renewal. Thus, in the absence of PKCθ, there is a significant increase in the quiescent SC pool following acute muscle injury. Overall, our findings suggest that, targeting PKCθ might be a potential novel approach to manipulate SC self-renewal in pathological conditions and improve regeneration.

## 2. Results

### 2.1. Phospho-PKCθ is Localized to the Nucleus, Centrosomes, Mitotic Spindle and Midbody in Satellite Cells during Mitosis

Satellite cells can exist in a quiescent state in physiological conditions, or as activated/committed progenitors during muscle regeneration [14]. To elucidate the functional role of PKCθ in satellite cells, we first analyzed the kinetics of PKCθ expression by Western blot during SC activation, proliferation and differentiation. Satellite cells isolated from hind limb muscle were processed immediately after isolation in order to analyze PKCθ expression in early activated cells [15]. A portion of the isolated cells were cultured in growth medium (GM) for 72h to analyze PKCθ expression in proliferating cells. For the analysis of PKCθ in differentiated cells, SCs were cultured in differentiation medium (DM) for 24h. Western Blot analysis showed that PKCθ is already expressed in early activated cells (i.e., freshly isolated). Its expression declines in proliferating cells and increases again in differentiated cells (Figure 1A,B). However, the analysis of the phosphorylated/active form revealed that PKCθlevel of activation is very similar between Freshly Isolated SCs and proliferating SCs (72h GM), while it is reduced in differentiating cells (24h DM), although not significantly (Figure 1A,C).

We then analyzed the localization of active PKCθ by immunofluorescence analysis of phospho-PKCθ in SCs isolated from WT mice and cultured for 24 and 72h.To identify proliferating SCs and visualize the mitotic spindle we performed α-Tubulin co-immunostaining. At 24h in culture, when SCs did not enter the cell cycle yet, phospho-PKCθ was localized on plasma membrane (Figure 1D). In proliferating cells, at 72h in culture, phospho-PKCθ was localized in the nucleus during prophase and in the centrosomes and partially in the mitotic spindle during metaphase. During telophase, phospho-PKCθ was visible at the spindle midzone, and during cytokinesis it moved to the intercellular bridge of the midbody (Figure 1D).

These results suggest that PKCθ could be involved in the regulation of SCs division processes.

### 2.2. Knock out of PKCθ Does Not Affect Satellite Cell Proliferation

To understand whether SC proliferation is affected by PKCθ,we isolated SCsfrom WT and PKCθ-/- muscles, labeled them with the CFSE proliferation die, and cultured in growth medium for 72h.FACS analysis showed that there was no difference in CFSE fluorescence between WT and PKCθ-/- cells (Figure 2A), suggesting that the cells underwent similar rounds of cycling. SC proliferation was also analyzed in regenerating gastrocnemius (GA) muscles by immunofluorescence staining for Pax7 as a SC marker, and Ki67 as a proliferation marker. The analysis was performed at day 3 and 7 after CTX injury, during the peak of SC proliferation and the initial phase of regeneration, respectively. The number of cells double positive for Pax7 and Ki67 were counted, and the results were normalized to the total number of Pax7^+^ cells. As shown in Figure 2B,C, the percentage of proliferating SCs (Pax7^+^/Ki67^+^) was similar between WT and PKCθ-/- mice.

### 2.3. Lack of PKCθ Stimulates Symmetric Self-Renewal by Regulating Pard3 Polarization

Together, the results obtained demonstrate that satellite cell proliferation is not regulated by PKCθ; thus, the localization of phospho-PKCθ at the centrosomes and mitotic spindle of dividing SC should reflect a role of this protein to other processes associated to cell division, rather than proliferation. Interestingly, in T cells, PKCθ promotes MTOC reorientation and cell polarization in the direction of the antigen presenting cell. This process may also promote asymmetric division, that is important for the acquisition of T cell memory [16]. We thus hypothesized that PKCθmay be involved in the regulation of symmetric/asymmetric division in SCs.

To investigate this possibility, we isolated single myofibers from WT and PKCθ-/- EDL muscles, and cultured them for 48h, which is the time necessary for the first cell division to occur. In this system SCs remain associated with the fiber, allowing us to study their behavior in conditions similar to their in vivo niche environment. During in vitro culture of myofibers SCs become activated, upregulate MyoD and enter cell cycle. Symmetric division generates two identical cells, which can be both Pax7^+^/MyoD^−^, or both Pax7^+^/MyoD^+^. Asymmetric division generates two cells that differ for MyoD expression, instead: one cell will be Pax7^+^/MyoD^−^, and the other will be Pax7^+^/MyoD^+^. After 48h in culture cell doublets were visible, and symmetric/asymmetric division was studied by immunofluorescence staining for Pax7 and MyoD.

The SCs destined to undergo self-renewal were identified as Pax7^+^MyoD^−^, whereas SCs committed to differentiation were Pax7^+^MyoD^+^. We observed a significant increase in the number of symmetric SC divisions in myofibers isolated from PKCθ-/- mice, compared to WT (Figure 3A,B). Within the symmetric division events, the number of “self-renewing” Pax7^+^/MyoD^−^ cell doublets was significantly higher in myofibers from PKCθ-/- mice compared to WT (Figure 3C).

The number of Pax7^+^/MyoD^−^ self-renewing cells remained higher in PKCθ-/- myofibers even after 72h in culture, when clusters of proliferating SCs were visible on the myofiber surface (Appendix A). The number of satellite cells per cluster was similar between the two genotypes, confirming that PKCθ does not affect satellite cell proliferation (Appendix A).

Asymmetric cell division is regulated by the polarization of PAR complex, which drives the segregation of polarity proteins that determine the fate of daughter cells [17].

Therefore, we wondered whether the increased symmetric divisions observed in myofiber associated SCs from PKCθ-/- mice may depend on an alteration of PAR3 complex polarization. To investigate this possibility, we examined the expression of Pard3 (the mammal ortholog of Drosophila Par3) and Pax7 on single myofibers isolated from WT and PKCθ-/- mice, after 36h in culture, by immunofluorescence staining. At this time point, polarity proteins are segregated, but the first mitotic division has not occurred yet. In Both WT and PKCθ-/- myofibers, 66% of the SCs showed low to absent Pard3 expression. Among the Pard3 expressing cells, we found a significant increase in the percentage of cells showing symmetric distribution of Pard3 in PKCθ-/- compared to WT SCs (Figure 3E–G). Likewise, there was a reduction in the frequency of cells showing asymmetric distribution of Pard3 in PKCθ-/-, compared to WT SCs (Figure 3E–G).

Together, these results suggest that PKCθ regulates SC self-renewal in vitro, and its absence promotes symmetric cell division.

### 2.4. Pharmacological Inhibition of PKCθ Increases the Fraction of Reserve Cell Population in Cultured Primary Myoblasts

We next sought to validate our results from the PKCθ-/- mouse model, using the C20specific inhibitor of PKCθ [11,12,18,19]. To analyze the effects of C20 throughout the different phases of myogenic progression, we isolated SC from WT and PKCθ-/- mice and cultured them for 4 days in growth medium (GM), followed by 2 days in differentiation medium (DM).C20 was used at the concentration range (0.5 µM, 1 µM and 2 µM) we previously found not to be toxic for in vitro treatment [11]. Control cultures were treated with 0.1% DMSO, the same concentration used for C20 dilution. To analyze any possible indirect effect of C20, we treated also PKCθ-/- SCs with C20, at the maximum concentration used for the experiment (2 µM).

Treatment of WT SCs with C20 significantly increased the fraction of Pax7^+^/MyoD^−^ reserve cells in a dose dependent manner (Figure 4A,B, Appendix A). In parallel, the fraction of cells committed to differentiation Pax7^−^/MyoD^+^ was reduced (Figure 4C,D, Appendix A). The fusion index was also reduced after treatment with C20, but the reduction was significant only at the highest dose of C20 used (Figure 4E,F, Appendix A). Similar results were obtained by analyzing the phenotype of the PKCθ-/- cultured cells (Figure 4 A,C,E), and C20 treatment did not exert any effect (not shown). These results suggest that in vitro pharmacological inhibition of PKCθ with C20 increases the fraction of ‘self-renewing’ SCs.

### 2.5. PKCθ Absence/Inhibition Increases the Quiescent Satellite Cell Pool after Induction of Acute Injury

Since the results obtained on myofibers in vitro suggested that PKCθ may regulate SCs self-renewal, we examined whether PKCθ controls the extent of the SCs pool in vivo.

To study SCs self-renewal in vivowe first analyzed the number of SCs in WT and PKCθ-/- mice at 7 and 28 days after cardiotoxin (CTX) muscle injury, when the muscle is regenerating or is completely regenerated, respectively. Contralateral uninjured muscle was used as control.

Immunofluorescence analysis of Pax7^+^ cells revealed that the number of SCs per mm^2^ and the number of SCs per fiber was similar in PKCθ-/- and WT gastrocnemius (GA) uninjured muscles (Figure 5B,C, Appendix A). At day 7 after injury, the number of Pax7+ cells was increased in both WT and PKCθ-/- mice, as a result of cell proliferation. However, the number of Pax7+ cells in PKCθ-/- mice was significantly higher compared to WT mice (Appendix A). At day 28 after CTX injury, when muscle is completely regenerated and SCs have returned to quiescence, the number of Pax7^+^ cells was significantly higher in PKCθ-/- muscle compared to WT, with a64.4% increase (Figure 5A–C). To confirm that at this stage all the SCs have gone back to quiescence, we analysed their cycling status by immunofluorescence staining for Pax7 and Ki67. The results showed that more than 99% of the Pax7^+^ cells were negative for Ki67 in both WT and PKCθ-/- mice, indicating that they are not proliferating (Figure 5F). Moreover, all the cells analyzed 28days after CTX were localized in their final position as quiescent cells, beneath the basal lamina and the sarcolemma of muscle fibers (Figure 5A).

These results suggest that the pool of quiescent SCs is increased in the absence of PKCθ-/- following injury.

To compare the regenerative ability of WT and PKCθ-/- mice, we analyzed myofiber CSA 28 days after injury (Figure 5D,E): the mean myofiber CSA, and the distribution of myofiber CSAs were similar in WT and PKCθ-/- mice. These results suggest that lack of PKCθ increases SC self-renewal without affecting the muscle regenerative ability after injury.

To investigate whether pharmacological inhibition of PKCθ leads to similar results, we treated WT mice with the pharmacological inhibitor of PKCθ (C20) and analyzed theSC number before and after CTX injury.

The treatment was started one day prior to the CTX injection, and was continued for 10 days following injury, during the phase of satellite cell activation, proliferation and differentiation (Figure 5G). The mice were treated with daily intra peritoneal injections of C20, at a dose of 5 mg/Kg (previously established to be effective in vivo [12]). Control mice were treated with Vehicle (DMSO) at the same concentration used to dissolve C20 (0.5% final concentration in the injected volume). Satellite cell number was analyzed by Pax7 immunofluorescence staining at day 28 after injury. The results showed that the number of Pax7^+^ cells per mm^2^, and the number of Pax7^+^ cells per fiber was similar in uninjured GA in C20 and vehicle treated mice. However, the number of SCs that returned to quiescence 28 days after injury, was significantly higher in mice treated with C20, compared to vehicle, with a 50% increase (Figure 5H,I). The analysis of the mean CSA and the CSA distribution of myofibers showed no significant differences between C20 and vehicle treated mice, indicating that C20 treatment does not affect the muscle regenerative ability (Figure 5J,K). These results suggest that pharmacological inhibition of PKCθ promotes SC self-renewal in vivo.

### 2.6. The Number of Quiescent Satellite Cells Increases in PKCθ-/- Mice after Repeated Injuries

Previous studies have shown that SC self-renewal is necessary to maintain the stem cell population after repeated traumas [20]. To investigate the behavior of SCs following repeated injury in the absence of PKCθ-/-, we induced three CTX injuries in WT and PKCθ-/- GA muscles, 20 days apart from each other. The muscles were analyzed 30 days after each injury, when regeneration is completed (Figure 6A). At one month after the first injury we observed a 64% increase in the number of SCs/mm^2^inPKCθ-/- mice compared to WT. After 2 injuries this increase remained constant (Figure 6C). However, one month after the third injury, while SC number did not change in WT mice, we observed an increase of 110% in the number of SCs/mm^2^ in PKCθ-/- mice compared to WT mice (Figure 6C). Moreover, the number of SCs per fiber was significantly higher in PKCθ-/- mice compared to WT at all time points analyzed after injuries.

The CSA of regenerated muscle fibers 30 days after the third injury was similar between WT and PKCθ-/- mice (Figure 6E,F), suggesting that, although more satellite cells undergo self-renewal in PKCθ-/- mice, their myogenic potential is maintained.

### 2.7. Knock-out of PKCθ Does Not Alter the Inflammatory Milieu after Induction of Acute Injury

Neutrophils and monocytes are the principal immune cells recruited to muscle following acute injury, where they influence SC behavior by stimulating their activation, proliferation and differentiation [21]. While PKCθ plays an important role in effector T cell recruitment in the context of chronic muscle inflammation as we reported previously [12], less is known about its role in myeloid cell recruitment. To understand whether PKCθ can indirectly affect SC behavior by altering myeloid cell response during acute injury, we analyzed myeloid cell infiltration in WT and PKCθ-/- muscle 3 and 10 days after CTX injury, by cytofluorimetric analysis. At 3 days after injury, inflammatory monocytes invade the injured muscle and drive satellite cell activation and proliferation. At 10 days after injury, the switch of the pro-inflammatory M1 to the anti-inflammatory M2 macrophages stimulates satellite cell differentiation and muscle regeneration. We found no significant difference in myeloid cell infiltration between WT and PKCθ-/- mice at 3 days after injury (Figure 7A–C). The total number of infiltrating mononuclear cells, CD45+ cells, and CD11b+ cells, was similar between the two genotypes. The M1 and M2 macrophages were identified as CD11b+ F4/80+ Ly6c-hi cells, and CD11b+ F4/80+ Ly6c-lo cells, respectively. As shown in (Figure 7D,E), the total number of M1 and M2 macrophages did not change significantly between WT and PKCθ-/- muscles. Within the M2 macrophage population, we analyzed the expression of CD206 marker, which was also similar (Figure 7F).

At 10 days after CTX injury, the total number of infiltrating inflammatory cells decreased to a similar extent in both WT and PKCθ-/- mice indicating a resolution of the inflammatory phase. At this stage, the total number of mononuclear cells, CD45+ cells, CD11b+ cells, M1 and M2 macrophages was similar between WT and PKCθ-/- mice (Figure 7G–K). There was no difference in the expression of the M2 macrophage marker CD206 (Figure 7L).

Pro-inflammatory cytokines induce SC activation and differentiation. Therefore, we analyzed the expression level of some of the principal inflammatory cytokines known to be released during muscle injury. The results obtained showed no significant difference in the expression level of TNF-α and IL-6 between WT and PKCθ-/- mice, 3 and 10 days after CTX injury (Appendix A). Altogether, these results suggest that PKCθ does not affect the inflammatory milieu during acute injury.

## 3. Discussion

During muscle regeneration a proportion of SCs undergo self-renewal in order to maintain the stem cell pool and preserve the regenerative capacity of muscle. The mechanism regulating this process is complex and involves the activity of diverse molecular pathways [2].

In the present study, we demonstrate that PKCθ plays an important role in SC self-renewal. PKCθ was expressed in activated and differentiating SCs with minimal expression in proliferating cells. Interestingly, the ratio between the active phospho-PKCθ and total PKCθ was stable in activated and proliferating cells, suggesting that even though PKCθ expression decreases during proliferation, its activity is maintained. In proliferating SCs, during prophase phospho-PKCθ was localized in the nucleus. A previous study carried out in T cells also showed that PKCθ is localized at the chromatin level [22]. However, ours is the first study to visualize PKCθ in association with DNA during prophase. Together, these observations may suggest that PKCθ is involved in the process of chromatin condensation; however, this possibility should be further investigated in the future. On the other hand, although we found that PKCθ localizes to the mitotic machinery, we did not observe any aberrant cell division in PKCθ-/- satellite cells, nor in WT cells treated with a PKCθ inhibitor, suggesting that PKCθ is not necessary for successful cytokinesis.

Using the murine erythroleukemia (MEL) cell line it was previously shown that PKCθ is recruited to the mitotic spindle and associates with centrosomes and kinetochore of dividing cells. Furthermore, in non-proliferating or differentiated MEL cells, PKCθ expression was down-regulated [23]. Although the effect of PKCθ ablation/inhibition on MEL cell proliferation was not analyzed in that study, a potential role of PKCθ in the regulation of cell growth and proliferation was proposed. In our study, the localization of phospho-PKCθ to the mitotic machinery in SC also suggested a role of this protein in the regulation of SC division. However, we showed that SC proliferation was not affected by the absence of PKCθin vitro as well as in vivo after CTX-induced muscle injury.

Since we detected phospho-PKCθ to the centrosomes and mitotic spindle of SCs, we reasoned that PKCθ might regulate SC polarity during division. Par-3, Par-6, and aPKC form a complex that localizes to the apical pole of the dividing stem cells, and distribute polarity proteins that direct mitotic spindle orientation [4,17]. We found that lack of PKCθ increases the symmetric distribution of Pard3 protein in dividing satellite cells, thereby increasing the number of symmetric divisions. As a result, we observed that both the genetic ablation and the pharmacological inhibition of PKCθ led to the expansion of the Pax7^+^/MyoD^−^ ‘reserve cell’ population, in vitro and in vivo. After the induction of muscle injury, the number of SCs going back to quiescence increased by 65% in PKCθ-/- mice, compared to WT, and pharmacological inhibition of PKCθ with C20 inhibitor led to similar results after injury. Notably, a 10-day treatment with C20 during the first phase of regeneration was sufficient to observe the increase of the quiescent SC pool, suggesting that a PKCθ inhibitor treatment can be useful even when administered for a short time. No alteration in immune response has been observed in PKCθ-/- regenerating muscle compared to WT, further supporting the notion that PKCθ plays an intrinsic role in SC function, through the regulation of polarity proteins distribution, as we propose.

Interestingly, it has been previously shown that, in mdx mice, the absence of dystrophin in SCs leads to the downregulation of the important regulator of Pard3 polarization, Mark2 (Par1b in Drosophila), since it confines Pard3 to the opposite side of the cell. In the absence of dystrophin, Mark2 downregulation causes a reduction of Pard3 polarization, and the number of satellite cell symmetric divisions generating self-renewing cells increases [24]. Furthermore, Watkins et al. demonstrated that nPKCs regulate Par1 activation and localization in HEK 293 cells, through PKD mediated phosphorylation [25]. PKCθ belongs to the family of nPKCs, and it is known to be an upstream regulator of PKD [26]. Thus, we could speculate that PKCθ regulates Par3 polarization via Par1 phosphorylation, but this aspect should be further investigated.

Overall, our findings suggest that PKCθ might be a promising target in conditions where SCs show functional decline and enhanced myogenic commitment, such as aging. The functional decline in aged satellite cells is due to modifications in the satellite cell niche, such as deregulations in Wnt, Notch, and TGFβ pathways [27]. Intrinsic defects in satellite cells from aged mice involve the upregulation of p-38 and JAK-STAT pathway, which results in the loss of self-renewal ability and regeneration. Treatment with p-38 inhibitors or JAK-STAT inhibitors leads to the rescue of self-renewal ability and improved regeneration in aged mice [28,29]. Therefore, future studies should investigate whether PKCθ expression/phosphorylation increases in aged satellite cells, and whether its inhibition can rescue self-renewal.

In conclusion, our results identify PKCθ as an important regulator of SC self-renewal and a novel potential target to counteract satellite cell loss in pathological conditions by limiting the number of cells committed to differentiation.

## 4. Materials and Methods

### 4.1. Animal Models

PKCθ-/- mice (C57BL/6J background) were originally provided by Dan Littman (New York University, New York, NY, USA). In these mice, the gene encoding PKCθ Is inactivated in all cells of the body, as previously described [30]. C57BL/6J control mice were purchased from Jackson laboratory (Bar Harbor, ME, USA). The animals were housed in the Histology Department–accredited animal facility. All the procedures were approved by the Italian Ministry for Health and were conducted accordingto the U.S. National Institutes of Health (NIH) guidelines (Approval number 60/2018-PR, 29/01/2018).

### 4.2. Cardiotoxin Injury

Cardiotoxin injury was performed by injecting 20 μL of Cardiotoxin from Naja Pallida (10 mMol in H2O, Latoxan (ZA Les Auréats, France), with two injections of 10 μL each, in two different areas of the GA muscle, using a 30 Gauge micro-syringe.

### 4.3. Single Myofiber Isolation and Culture

Single myofibers were isolated from EDL muscles of 4–6-week-old mice, as described [31,32,33]. Briefly, EDL muscles were dissected from tendon to tendon and incubated in DMEM (Sigma-Aldrich St. Louis, MS, USA) containing 0.2% collagenase I (Sigma-Aldrich) for 45 min. Myofibers were then dissociated by gentle flushing with a glass pipette, in a dish containing DMEM 1% penicillin streptomycin (Sigma-Aldrich). Then, myofibers were washed 3 times in DMEM 1% penicillin-streptomycin and left for 1h in the last washing dish. Myofibers were then cultured in suspension in Horse Serum (Gibco, Carlsbad, CA, USA) coated dishes, in high glucose DMEM containing 1% penicillin-streptomycin, 1% Chick Embryo Extract (CEE, produced in the lab), 20% FBS (Sigma-Aldrich).

### 4.4. Satellite Cell Isolation and Culture

SCs were prepared from hind-limb muscles of 4–8 week-old mice, as previously described [34]. Briefly, Hind limb muscles were dissected with scissors and finely diced with a scalpel. For enzymatic digestion, muscles were incubated in Collagenase type II (Sigma-Aldrich) 0.4 mg/mL in PBS (Sigma-Aldrich), for 45 min in a shaking water bath at 37 °C. A second digestion was performed in1 mg/mL of Collagen/Dispase (Roche, Basel, Switzerland) in PBS Calcium-Magnesium free (Sigma-Aldrich), for 30 min at 37 °C. The digested muscle was then passed through 70 µm cell strainer first, and 40 µm cell strainer then, to remove debris. Next, satellite cell purification was performed by using SC Isolation Kit (Miltenyi Biotech, Bergisch Gladbach, Germany). Cells were then counted, washed, resuspended in growth medium (GM) and plated. GM contained DMEM20% HS, 3% Embryo Extract. Differentiation medium (DM) contained DMEM 5% HS, 1% CEE.

### 4.5. Immunofluorescence Analysis

Immunofluorescence analysis was performed as previously described [33]. For immunofluorescence on sections, 8 µm muscle cryosections were fixed in 4% PFA for 10 min at room temperature (RT), and then permeabilized in cold methanol for 6 min at –20 °C. Next, antigen retrieval was performed in Citric Acid 0.01M pH 6, for 10 min at 90 °C. Sections were then blocked in BSA (Sigma-Aldrich) 4%, Goat Serum (Sigma-Aldrich) 5% for 30 min RT. Sections were incubated with primary mouse anti-Pax7 antibody (1:10 Developmental Studies Hybridoma Bank, Iowa City, IA, USA) and rabbit anti-laminin (1:200 Sigma-Aldrich) diluted in BSA 4% over night at 4 °C. The next day, section were incubated with biotin anti-mouse (1:1000 BioLegend, San Diego, CA, USA) 1h RT. The sections were then washed three times with PBS for 15 min and subsequently incubated with secondary antibodies Streptavidin-Cy3 (1:500 BioLegend) and goat anti rabbit IgG Alexa Fluor 488 (1:1000 Abcam, Cambridge, UK). Nuclei were counterstained with Hoechst.

For myofiber staining, myofibers were fixed in PFA 4% for 5 min RT, permeabilized in cold methanol −20 °C for 6 min, and then incubated in Glycine 1M for 10 min RT to reduce background. Blocking was performed in BSA 4%, Goat Serum 5% for 30 min RT. Primary antibodies mouse anti Pax7, rabbit anti-MyoD (1:50 Santa Cruz C20: sc-304, Dallas, TX, USA), and rabbit anti Pard3 (07-330 Merck Millipore, Burlington, MA, USA) were incubated O/N at 4 °C. The next day myofibers were incubated for 1h at RT in biotin anti-mouse and subsequently in secondary antibodies goat anti rabbit Alexa Fluor 488 and Streptavidin Cy3 for 1h RT, at the concentration listed before. Nuclei were counterstained with Hoechst or Topro3. Samples were analyzed under an epifluorescence Zeiss Axioskop 2 Plus microscope (Carl Zeiss, Oberkochen, Germany).

### 4.6. Western Blot

Satellite cells were homogenized in ice-cold buffer containing 20 mMTris (pH 7.5), 2 mM EDTA, 2 mM EGTA, 250 mM sucrose, 5 mMDTT, 200 mg/mL leupeptin, 10 mg/mL Aprotinin, 1 mM PMSF, and0.1% Triton X-100 (all from Sigma-Aldrich) and then disrupted by sonication, as previously described [13]. The homogenate was then incubated for 30 min at 4 °C in rotation, then centrifuged at 12,000g for 10 min at 4 °C. Protein determination was performed by using the Comassie Plus protein assay reagent (Pierce, Rockford, IL, USA), according to the manufacturer’s instruction. Proteins from each sample were loaded onto 10% SDS-polyacrylamide gels and transferred to a nitrocellulose membrane (Schleicher and Schuell, Dassel, Germany). The membranes were incubated with anti-PKCθ (#2059, Cell Signaling, Danvers, MA, USA) and anti phospho-PKCθ (Thr538-Cell Signaling #9377) primary antibodies 1:1000 diluted in BSA 5% in TBS 1% tween. HRP-conjugated goat anti–rabbit IgG 1:5000 diluted in BSA 5% in TBS 1% tween (A120-101P, Bethyl Laboratories, Montgomery, TX, USA) was used as secondary antibody, and immunoreactive bands were detected using ECL solution, according to the manufacturer’s instructions. Chemiluminescent signals were acquired by ChemiDoc MP Imaging System (Bio-Rad, Hercules, CA, USA). Densitometric analysis was performed using ImageJ software (U. S. National Institutes of Health, Bethesda, MD, USA). Proteins were normalized to GAPDH (Santa Cruz 6C5: sc-32233).

### 4.7. Cell Isolation and Cytofluorimetric Analysis

To prepare single cell suspension, muscles were dissected from mice and finely minced with a scalpel in a dish containing DMEM, as previously described [12]. Next, minced muscles were digested in a solution of Collagenase type IV (Worthington, Lakewood, NJ, USA) 1 mg/mLin DMEM, (10 mL/g of muscle) for 1h 30 min in shaking water bath at 37 °C. Then, digested muscles were passed through a 70 µm cell strainer first and then through a 40 µm cell strainer, to exclude cell debris. Muscle single cell suspension was then resuspended in FACS buffer (PBS 1% FBS) and incubated 30 min on ice with the following antibodies: anti-CD45 (Biolegend, clone 30-F11, 1:6000), anti-Ly6G (eBioscience, clone 1A8 and clone RB6-8C5, 1:3000), anti-Ly6c (Biolegend, clone HK1.4, 1:100), anti-F4/80 (Biolegend, clone BM8, 1:1000), anti-CD206 (MMR) (Biolegend, clone C068C2, 1:50), anti-CD11b (Biolegend, clone M1/70, 1:3000). Cell viability was assessed with 4′,6-diamidino-2-phenylindole dilactate (DAPI) (BioLegend).

For CFSE analysis isolated SCs were stained with CFSE (ThermoFisher Scientific, Waltham, MA, USA) 5 µm for 20′ at 37 °C in dark prior to culture. Samples were processed using a Dako CyAn ADP flow cytometer and acquired data were analyzed using FlowJo software version 10 (FlowJo LLC, Ashland, OR, USA).

### 4.8. RNA Isolation and Real Time PCR Analysis

For RNA preparation from muscle, the muscles were homogenized in TissueLyser (Quiagen, Hilden, Germany) in the presence of TRI reagent (Sigma-Aldrich). Samples were then processed as previously described [11]. The following primers were used for amplification: IL-6 for: ATGAAGTTCCTCTCTGCAAGAGACT, rev: CACTAGGTTTGCCGAGTAGATCTC; TNF-α for: ATGATCCGCGACGTGAA, rev: AGGGAGGCCATTTGGGAA; GAPDH for: ACCCAGAAGACTGTGGATGG, rev: CACATTGGGGGTAGGAACAC. All Real Time PCR results are expressed as relative ratios of the target cDNA transcripts to GAPDH and normalized to that of the reference condition. Data analysis was performed using 7500 Software v2.0.6 provided by Applied Biosystems (Foster City, CA, USA). Data are expressed as fold-change in expression levels.

### 4.9. Statistical Analysis

All statistical analyses were performed using GraphPad Prism software version 6 (La Jolla, CA, USA). Quantitative data are presented as means ± SEM of at least three experiments. Statistical analysis to determine significance was performed using Student’s *t* tests or Anova test. Differences were considered statistically significant at the *p* < 0.05 level.

## Figures and Tables

**Figure 1 ijms-21-02419-f001:**
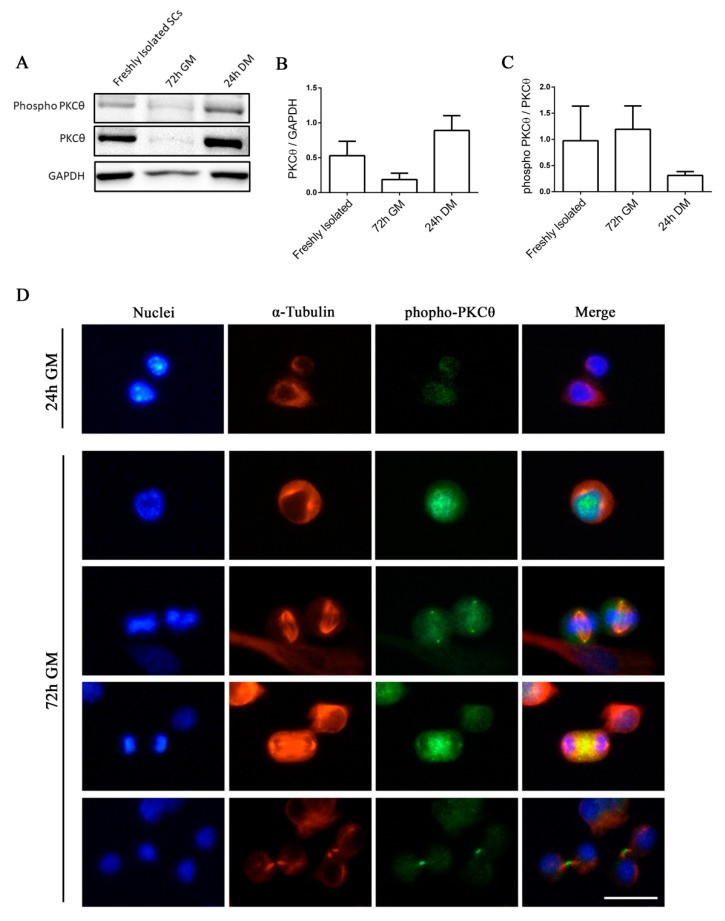
Phospho-PKCθ is localized to the nucleus, centrosomes, mitotic spindle and midbody in satellite cells during mitosis. (**A**): WB analysis showing PKCθ and phospho-PKCθ expression in activated SCs (freshly isolated), proliferating SCs (72h GM), differentiated myotubes (24h DM) (*n*= 3 individual experiments). (**B**): WB densitometric analysis of the level of PKCθ expression, normalized on the level of GAPDH protein. (**C**): WB Densitometric analysis of the phospho-PKCθ/PKCθ ratio. (**D**): Representative immunofluorescence images of satellite cells after 24h and 72h in culture, stained for α-Tubulin (red) and phospho-PKCθ (green). Nuclei were counterstained with Hoechst. Scale bar 20 µm. Error bars represent mean ± sem.

**Figure 2 ijms-21-02419-f002:**
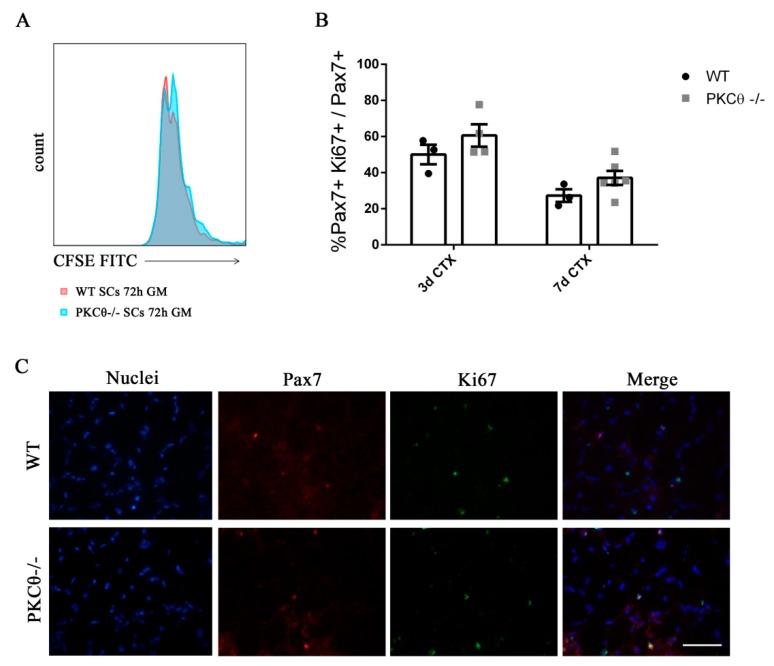
Knock out of Protein kinase C Theta (PKCθ) does not affect satellite cell (SC) proliferation. (**A**): Flow-cytometry analysis of CFSE stained WT and PKCθ-/- SCs, cultured for 72h in GM. (**B**): Percentage of proliferating SCs, identified by immunofluorescence asPax7^+^/Ki67^+^ cells on WT and PKCθ-/- GA muscle sections, 3 and 7 days after CTX injury, over the total Pax7^+^ cells. (**C**): Representative immunofluorescence images of WT and PKCθ-/- GA muscles, stained for Pax7 and Ki67, 3 days after the induction of CTX injury, scale bar: 100 µm.

**Figure 3 ijms-21-02419-f003:**
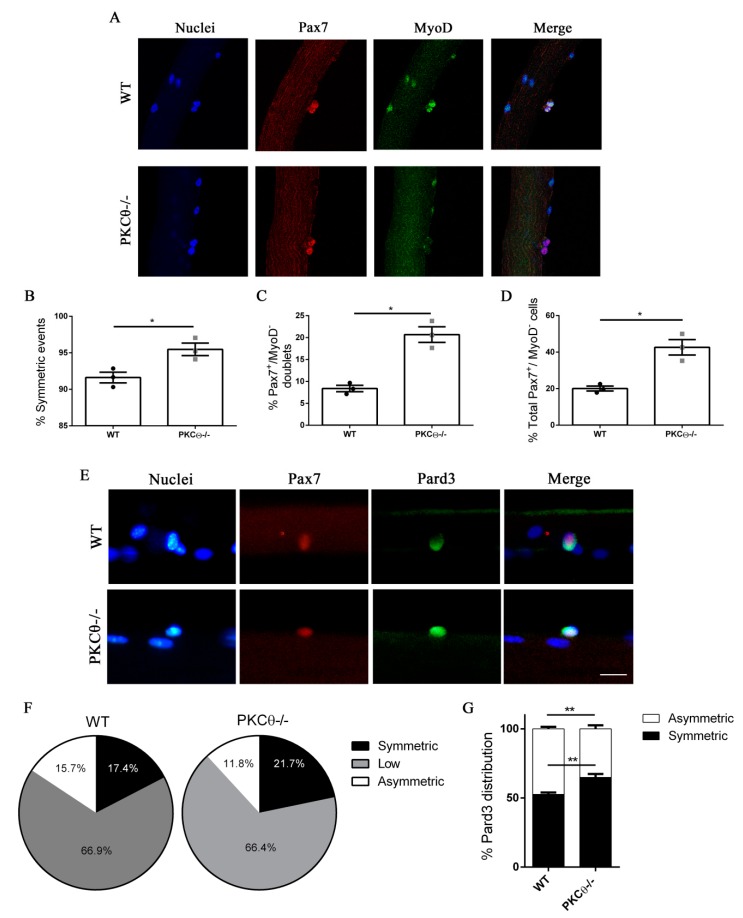
Lack of PKCθ stimulates symmetric self-renewal by regulating Pard3 polarization. (**A**): Representative pictures of single myofibers isolated from EDL muscles of WT and PKCθ-/- mice, after 48h in culture. Myofibers were stained for Pax7 (red) and MyoD (green), nuclei were counterstained with Topro3. (**B**): Quantification of symmetric division events. (**C**): Quantification of Pax7^+^/MyoD^−^ cell doublets, and (**D**): quantification of total Pax7^+^/MyoD^−^ cells in WT and PKCθ-/- single myofibers. (**E**): Representative pictures of single myofibers isolated from EDL muscles of WT and PKCθ-/- mice, after 36h in culture. Myofibers were stained for Pax7 (red) and Pard3 (green), nuclei were counterstained with Topro3. (**F**): Percentage ofSCs showing symmetric, low or asymmetric Pard3 distribution in WT and PKCθ-/- myofibers. (**G**): Percentage of SCs showing symmetric and asymmetric Pard3 distribution (WT, *n* = 3 mice, PKCθ-/-, *n* = 3 mice, *n* > 20 myofibers analyzed per mouse). Error bars represent mean ± sem, * *p* < 0.05 calculated by Student’s *t*-test.

**Figure 4 ijms-21-02419-f004:**
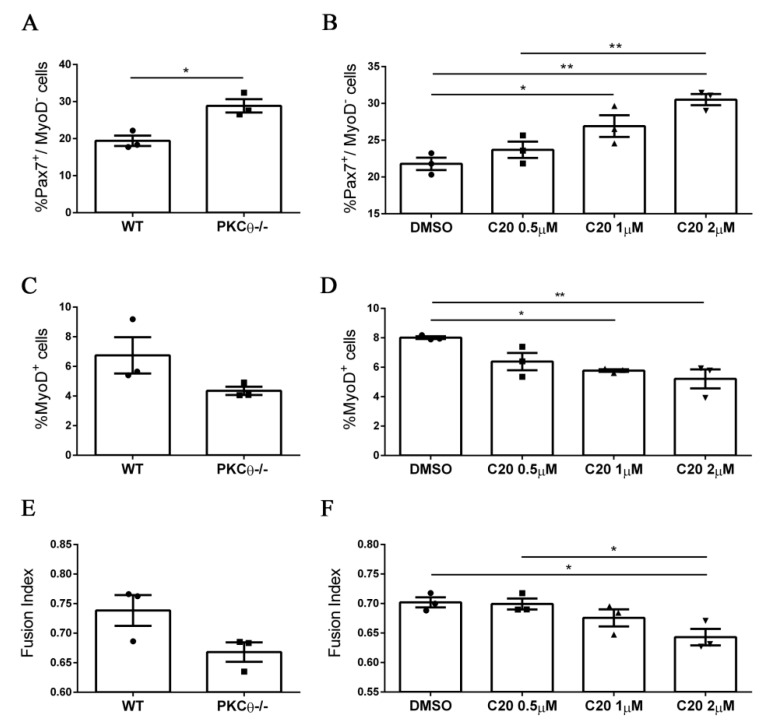
Pharmacological inhibition of PKCθ increases the fraction of reserve cell population in cultured primary myoblasts. (**A**): Percentage of Pax7^+^/MyoD^−^ SCs in WT and PKCθ-/- cultures, or in WT cultures treated with increasing concentration of C20 (**B**); cells were cultured for 4 days in GM followed by2 days in DM. (**C**): Percentage of total MyoD^+^ SCs in WT and PKCθ-/- cultures, or in WT cultures treated with C20, as in B (**D**). (**E**): Fusion index of WT and PKCθ-/- myotubes, or WT myotubes treated with C20 (**F**) as in A (*n* = 3 replicate dishes per group). Error bars represent mean ± sem, * *p* < 0.05, ** *p* < 0.01 calculated by one-way ANOVAwith adjustment for multiple comparison test.

**Figure 5 ijms-21-02419-f005:**
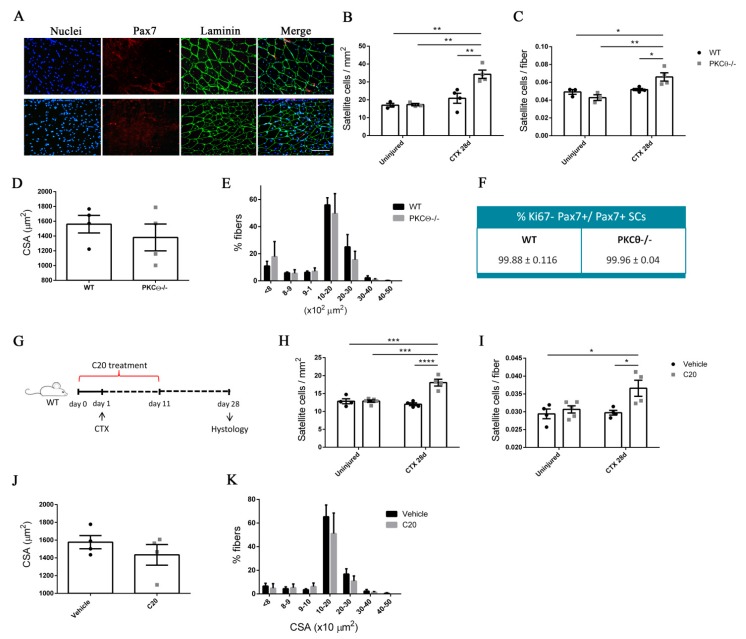
PKCθ absence/inhibition increases the quiescent satellite cell pool after induction of acute injury. (**A**): Representative immunofluorescence pictures of WT and PKCθ-/- GA sections, 28 days after CTX injury. Sections were stained for Pax7 (red) and Laminin (green). Nuclei were counterstained with Hoechst. Scale bar: 100 µm. (**B**): Number of SCs per mm2 and (**C**): number of SCs per fiber in uninjured and 28 day-injured GA muscle, in WT and PKCθ-/- mice. (**D**): Mean CSA and (**E**): CSA distribution of muscle fibers in WT and PKCθ-/- GA sections, 28 days after injury. (**F**): Quantification of non-proliferating SCs 28 days after CTX injury, in WT and PKCθ-/- GA, identified by immunofluorescence co-staining for Pax7 and Ki67. (WT, *n* = 4 mice, PKCθ-/-, *n* = 4 mice). (**G**): experimental plan for in vivo C20 treatment in injured muscle. (**H**): Number of SCs per mm^2^ and (**I**): number of SCs per fiber in uninjured and 28 day-injured GA muscle, in WT mice treated with C20 or vehicle. (**J**): mean CSA and (**K**): CSA distribution of muscle fibers in WT mice treated with C20 or vehicle, 28 days after injury. (C20 treated WT, *n* = 4 mice, Vehicle treated WT *n* = 4 mice). Error bars represent mean ± sem, * *p* < 0.05, ** *p* < 0.01 *** *p* < 0.001, **** *p* < 0.0001 calculated by Two-way Anova with adjustment for multiple comparison test.

**Figure 6 ijms-21-02419-f006:**
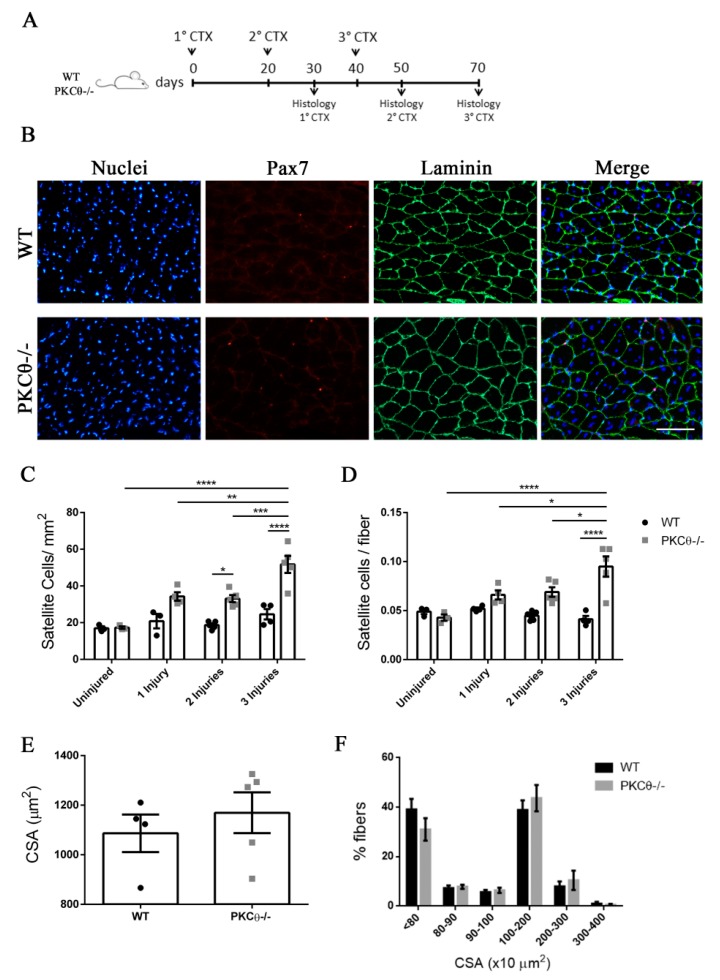
The number of quiescent satellite cells increases in PKCθ-/- mice after repeated injuries. (**A**): Experimental plan. (**B**): representative immunofluorescence pictures of WT and PKCθ-/- GA sections 30 days after 3 CTX injury, scale bar: 100 µm. Sections were stained for Pax7 (red) and Laminin (green). Nuclei were counterstained with Hoechst. (**C**): Quantification of the number of SCs/mm^2^ (**D**): and number of SCs per fiber in uninjured GA from WT and PKCθ-/- mice, or after 1, 2 and 3 injuries. (**E**): mean CSA of regenerated fibers 30 days after the third injury in WT and PKCθ-/- mice. (**F**): Frequency of myofiber CSA from GA muscles of WT and PKCθ-/- mice, 30 days after the third injury. Error bars represent mean ± sem, * *p* < 0.05, ** *p* < 0.01, *** *p* < 0.001, **** *p* < 0.0001 calculated by two-way ANOVA with adjustment for multiple comparison test.

**Figure 7 ijms-21-02419-f007:**
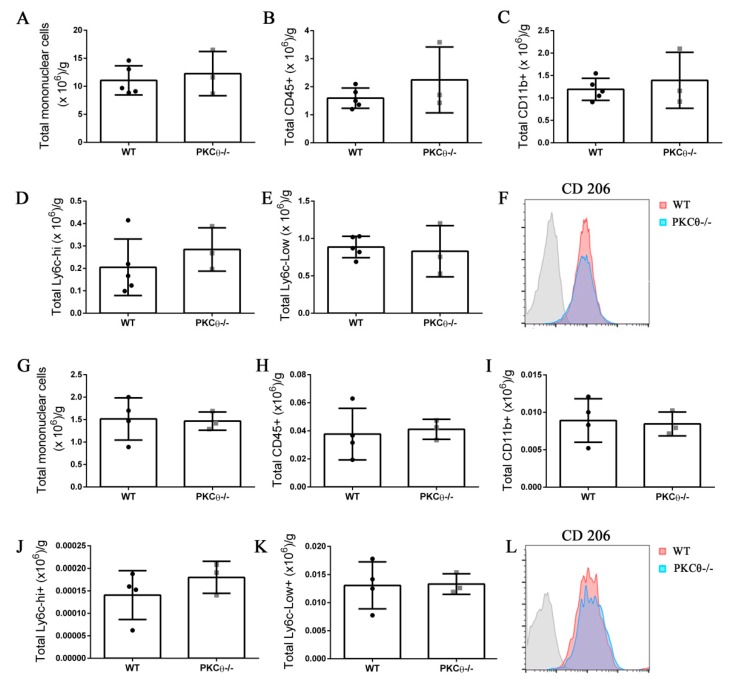
Knock-out of PKCθ does not alter the inflammatory milieu after induction of acute injury. (**A**): Total number of mononuclear cells, (**B**): CD45+ cells, (**C**): CD11b+ cells, (**D**): Ly6c-hi+ cells, and (**E**): Ly6c-low+ cells, normalized on muscle mass 3 days after CTX injury in WT and PKCθ-/- GA. (**F**): histogram showing CD206 mean fluorescence in WT and PKCθ-/- GA 3d after CTX. (**G**): Total number of mononuclear cells, (**H**): CD45+ cells, (**I**): CD11b+ cells, (**J**): Ly6c-hi+ cells, and (**K**): Ly6c-low+ cells, normalized on muscle mass 10 days after CTX injury in WT and PKCθ-/- GA. (**L**): histogram showing CD206 mean fluorescence in WT and PKCθ-/- GA 10d after CTX. Error bars represent mean ± sem.

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
