# Peer review of "Targeting PKCθ Promotes Satellite Cell Self-Renewal"

_ijms, 2020, doi:10.3390/ijms21072419_

Round 1
Reviewer 1 Report
This study aims at determining the role of PKCθ on SCs activity. Authors show that PKCθ plays a direct role on satellite cells activity by stimulating their symmetric division and promoting their self-renewal.
This very interesting work opens new avenues for improving muscle regeneration and during pathological situations.
There are some points that should be addressed by the authors.
Fig. 1:
1) When analyzing phosphorylation levels to assess how active or inactive a kinase is to induce a downstream signaling cascade, is it better to analyze the ratio of phosphorylated kinase/total kinase or to analyze the phosphorylated kinase only. This is especially true when the total kinase exhibits very variable expression levels from a condition to another. This is the case in this study, since the total PKCθ is well expressed in the early-activated cells, and then is drastically reduced in the proliferating cells before being again high expressed in differentiating cells. For instance, in this study the levels of phosphorylated PKCθ did not increase at 72h of incubation in growth medium. One may considered that the protein levels for phosphorylated PKCθ were stable between freshly isolated SCs and SCs in proliferation (as GAPDH also exhibit a slight decrease at 72GM compared to freshly isolated SCs or 24 DM). Authors should take this into consideration before concluding that the activity of PKCθ is high in proliferating SCs in the beginning of the discussion section.
2) Did authors have data about the localization of PKCθ in differentiated cells after 24h culture on DM ?
Fig. 3: Could authors precise how the distinction between symmetric and asymmetric SCs has been made methodologically? Was it only using Pax7+/MyoD- and Pax7+/MyoD+ cells. Because, Pax7/MyoD- cells can come from either symmetric or asymmetric division? Right?
Fig. 4: DMSO was used for solubilization of the pharmacological inhibitor of PKCq. However, there is no information about the % of DMSO used for solubilization and thus of the % of DMSO the cells are facing to. It seems that DMSO is usually well tolerated with no observable toxic effects to cells at 0.1% final concentration (v/v). At 1% or higher, toxic effects have been reported. DMSO cytotoxicity depend on the concentration, on the type of cell or cell line, and on the culture/incubation conditions: media, time etc. Did authors performed proper control experiments with a dose gradient of DMSO with their cells, under the exact same incubation conditions without the test compound, determining general cytotoxicity, e.g. with the MTT test? Same for in vivo experiments.
Fig. 5: Did authors know how the inhibition of PKCθ impacts the fiber CSA 7d post CTX?
Author Response
This study aims at determining the role of PKCθ on SCs activity. Authors show that PKCθ plays a direct role on satellite cells activity by stimulating their symmetric division and promoting their self-renewal.
This very interesting work opens new avenues for improving muscle regeneration and during pathological situations
We thank the reviewer for the positive comment
There are some points that should be addressed by the authors.
Fig. 1:
1) When analyzing phosphorylation levels to assess how active or inactive a kinase is to induce a downstream signaling cascade, is it better to analyze the ratio of phosphorylated kinase/total kinase or to analyze the phosphorylated kinase only. This is especially true when the total kinase exhibits very variable expression levels from a condition to another. This is the case in this study, since the total PKCθ is well expressed in the early-activated cells, and then is drastically reduced in the proliferating cells before being again high expressed in differentiating cells. For instance, in this study the levels of phosphorylated PKCθ did not increase at 72h of incubation in growth medium. One may considered that the protein levels for phosphorylated PKCθ were stable between freshly isolated SCs and SCs in proliferation (as GAPDH also exhibit a slight decrease at 72GM compared to freshly isolated SCs or 24 DM). Authors should take this into consideration before concluding that the activity of PKCθ is high in proliferating SCs in the beginning of the discussion section.
We agree with the Reviewer; we modified the WB analysis to show phosphor-PKCθ/PKCθ ratio, and replaced the graph in Figure1C, accordingly. Indeed, we found that PKCθ level of activation is very similar between Freshly Isolated SCs and proliferating SCs (72h GM), while it is reduced in differentiating cells (24h DM),although not significantly (Fig. 1A and C), as specified on Page 2 lines 73-75. The discussion was changed accordingly “Interestingly, the ratio between the active phospho-PKCθ and total PKCθ was stable in activated and proliferating cells, suggesting that even though PKCθ expression decreases during proliferation, its activity is maintained” Page 14 lines 312-314.
2) Did authors have data about the localization of PKCθ in differentiated cells after 24h culture on DM?
We thank the reviewer for this comment; our unpublished observation showed that phospho- PKCθlocalization is diffused throughout the cell membrane in differentiated cells. However, more detailed analyses are needed to further elucidate its localization and role in myotubes.
Fig. 3: Could authors precise how the distinction between symmetric and asymmetric SCs has been made methodologically? Was it only using Pax7+/MyoD- and Pax7+/MyoD+ cells. Because, Pax7/MyoD- cells can come from either symmetric or asymmetric division? Right?
We agree with the reviewer, and we further explained the way we made the distinction (page 5, lines 124-126). The distinction was made by analyzing Pax7 and MyoD expression by immunofluorescence, assuming that symmetric division generates two identical cells, which can be both Pax7+/MyoD-, or both Pax7+/MyoD+, while asymmetric division generates two cells that differ for MyoD expression, one cell will be Pax7+/MyoD-, and the other will be Pax7+/MyoD
Fig. 4: DMSO was used for solubilization of the pharmacological inhibitor of PKCq. However, there is no information about the % of DMSO used for solubilization and thus of the % of DMSO the cells are facing to. It seems that DMSO is usually well tolerated with no observable toxic effects to cells at 0.1% final concentration (v/v). At 1% or higher, toxic effects have been reported. DMSO cytotoxicity depend on the concentration, on the type of cell or cell line, and on the culture/incubation conditions: media, time etc. Did authors performed proper control experiments with a dose gradient of DMSO with their cells, under the exact same incubation conditions without the test compound, determining general cytotoxicity, e.g. with the MTT test? Same for in vivo experiments.
We apologize for this carelessness. We now added this information (page 7, line 172; page 8, line 222) In particular, the final concentration of DMSO used in our in vitro studies was 0.1% (2µl DMSO into 2000µl total volume) for all the concentrations of C20 used, and for the control cultures with DMSO only, which is known to be well tolerated. We did not perform an MTT test, however, we did not observe any significant phenotypic changes between the DMSO treated SCs and the non-treated SCs in terms of differentiation (fusion index).
For our in vivo experiments, the C20/DMSO solution was diluted in PBS and administered by i.p. injections, with a DMSO 0.5% of the final injection volume, which is considered very low and safe.
Fig. 5: Did authors know how the inhibition of PKCθ impacts the fiber CSA 7d post CTX?
We thank the reviewer for raising this question; as further described in the introduction, Page 2 lines 45-47, our lab previously demonstrated that PKCθ regulates muscle development and homeostasis: muscle regeneration is delayed in PKCθ-/- mice, and myofiber CSA is reduced 7 days after injury (Madaro et al. MBoC, 2011). However, one month after injury there is no difference between WT and PKCθ-/- mice in terms of myofiber CSA and muscle organization, suggesting that regeneration in PKCθ-/- mice is delayed but not compromised.
In the present study we show that lack of PKCθ increases the fraction of SCs undergoing self-renewal. This result could further explain the smaller CSA observed in PKCθ-/- mice at 7 days after injury.
Reviewer 2 Report
The authors characterize the function of PKC Theta in satellite cells and muscle regeneration. The experiments are well performed and the interpretation of the results is correct.
Minor concerns:
- Muscle regeneration is characterized at 28 days after injury. At this time point, compensation may have happened and some early phenotypical changes due to the effects described by lack of PKC Theta on satellite cells not detected. It would be nice if the authors can report if there are changes in fiber size at earlier time points (i.e. 5 to 7 days post injury).
- The sentence on page 5, line 144 is misleading. It is said that PKC Theta promotes symmetric cell divisions. Do the authors mean “lack of PKC Theta”? Please, clarify.
Author Response
The authors characterize the function of PKC Theta in satellite cells and muscle regeneration. The experiments are well performed and the interpretation of the results is correct.
We thank the reviewer for the positive comment
Minor concerns:
- Muscle regeneration is characterized at 28 days after injury. At this time point, compensation may have happened and some early phenotypical changes due to the effects described by lack of PKC Theta on satellite cells not detected. It would be nice if the authors can report if there are changes in fiber size at earlier time points (i.e. 5 to 7 days post injury).
We thank the reviewer for raising this question; as further described in the introduction, Page 2 lines 45-47, our lab previously demonstrated that PKCθ regulates muscle development and homeostasis: muscle regeneration is delayed in PKCθ-/- mice, and myofiber CSA is reduced 7 days after injury (Madaro et al. MBoC, 2011). However, one month after injury there is no difference between WT and PKCθ-/- mice in terms of myofiber CSA and muscle organization, suggesting that regeneration in PKCθ-/- mice is delayed but not compromised.
In the present study we show that lack of PKCθ increases the fraction of SCs undergoing self-renewal. This result could further explain the smaller CSA observed in PKCθ-/- mice at 7 days after injury.
- The sentence on page 5, line 144 is misleading. It is said that PKC Theta promotes symmetric cell divisions. Do the authors mean “lack of PKC Theta”? Please, clarify.
We thank the Reviewer for pointing out this error. We changed the text to “PKCθ absence promotes symmetric cell division”, page 5 lines 150-151.